# Comparative Study of Optical Markers to Assess Bait System Efficiency Concerning Vaccine Release in the Oral Cavity of Dogs

**DOI:** 10.3390/v13071382

**Published:** 2021-07-15

**Authors:** Anna Langguth, Kansuda Leelahapongsathon, Napasaporn Wannapong, Suwicha Kasemsuwan, Steffen Ortmann, Ad Vos, Michael Böer

**Affiliations:** 1University of Veterinary Medicine Hannover, Foundation, Bünteweg 2, 30559 Hannover, Germany; 2Faculty of Veterinary Medicine, Kasetsart University, Bangkok 10900, Thailand; fvetkul@ku.ac.th (K.L.); napasaporn.wan@ku.th (N.W.); fvetswk@ku.ac.th (S.K.); 3Ceva Innovation Center GmbH, Am Pharmapark, 06830 Dessau-Rosslau, Germany; steffen.ortmann@ceva.com (S.O.); ad.vos@ceva.com (A.V.); 4Department of Ethology, University of Osnabrück, Barbarastraße 11, 49076 Osnabrück, Germany; michael.boer@tiho-hannover.de; 5Institute for Terrestrial and Aquatic Wildlife Research, University of Veterinary Medicine Hannover, Foundation, Bünteweg 2, 30559 Hannover, Germany

**Keywords:** markers, dogs, oral vaccination, rabies

## Abstract

Oral vaccination of dogs against rabies has the potential to achieve mass coverage and thus deplete the virus of its most important reservoir host species. There is, however, no established non-invasive method to evaluate vaccine release in the oral cavity, following bait ingestion. In this study, two pre-selected marker methods in conjunction with their acceptance were assessed in local Thai dogs. Shelter dogs (*n* = 47) were offered one of four randomized bait formulations; bait type A-, containing Green S (E142) in a fructose solution; type B-, containing Patent Blue V (E131) in a fructose solution; type C-, containing the medium used for delivery of oral rabies vaccine in baits commercially produced; and type D-, containing denatonium benzoate, which was to serve as the negative control, due to its perceived bitterness. Patent Blue V was found to possess overall stronger dyeing capacities compared to Green S. Furthermore, there was no significant difference in the acceptance or bait handling of Patent Blue V baits compared to those containing the oral rabies vaccine medium alone, suggesting the potential use of this dye as a surrogate for rabies vaccine when testing newly developed bait formats.

## 1. Introduction

Over a century after the first successful vaccination of a human [1], rabies is still widely spread around the globe, claiming an estimated 60.000 human lives each year [2]. Due to its inevitably fatal course and the wide array of susceptible host species, the virus is considered to be one of the most, if not the most important viral zoonotic disease worldwide [3,4,5]. Dogs are the main reservoir species, responsible for 99% of human cases [6]. While canine rabies remains under control in most developed areas of the world due to ubiquitous vaccination regiments, among other things [7,8], these methods may not be feasible in many countries where the disease is still endemic. In addition to the difficulty of reaching free-roaming sub-populations of dogs, financial and social constraints often do not allow for employment of the necessary staff and equipment to execute mass vaccination campaigns via parenteral route [9,10]. According to the World Health Organization (WHO) and the World Organization for Animal Health (OIE), a minimum of 70% of the dog population in a specific area needs to be immunized against rabies to achieve a lasting protective herd immunity [11]. Compared to vaccination campaigns focusing solely on parenteral immunization, the use of oral vaccination baits has been shown to be more effective in terms of cost per dog vaccinated and apparent coverage achieved [10,12]. Oral rabies vaccination (ORV) in dogs has the potential to fulfil the criteria set by both the WHO and the OIE for control and eventual elimination of dog-mediated rabies [12,13]. This fact, plus the aforementioned financial incentive, makes ORV in dogs a promising method to achieve the goal of global elimination of dog-mediated human rabies deaths by 2030 [2].

The concept of ORV is based on 3 components: the vaccine, the bait, and the distribution system. Bait uptake is mainly monitored by the detection of a marker substance such as tetracycline incorporated into the bait matrix. Upon bait consumption, this marker is taken up by the animal and deposited in bone and teeth, where it can later be detected via fluorescence microscopy [14]. In addition to tetracycline, other markers have been used to assess bait ingestion, including various types of iophenoxic acids (which can be detected in blood samples), as well as clenbuterol and rhodamine B (which are detectable in both fur and whiskers) [15,16,17,18]. However, the vaccine bait should not only be highly attractive to the targeted species, but also ensure prompt release of its fluid contents into the oral cavity. If an animal swallows the bait without perforating the incorporated sachet, the vaccine will not come into contact with the tonsils or oral mucous membranes, where an immune response may be triggered [19]. Therefore, not all individuals that accept and consume a vaccine bait can automatically be considered vaccinated [20]. In other cases, animals may attempt to separate the bait matrix from the sachet, or rough handling of the bait may occur, both of which result in sometimes significant spillage and loss of inoculant [19]. To address these concerns and further optimize ORV delivery methods, bait studies should not only assess bait acceptance but also evaluate bait handling in detail.

This requires a reliable method for documenting the efficient release of the sachet contents in the oral cavity. The use of marker substances like iophenoxic acid, tetracycline, or rhodamine B, incorporated into the bait matrix, would not suffice, as these can also be taken up in the intestine. Thus, it is not possible to determine how and where exactly these substances were absorbed, should samples test positive. Although rhodamine B has been used for detection in the oral cavity [21], it is not readily visible from a distance. For some species, including free-roaming dogs, which are largely inaccessible, inspection of the oral cavity at close range is often not feasible. Consequently, this study’s aim was the selection of a suitable marker substance, used as a surrogate for the vaccine, which allows a reliable evaluation of a vaccine bait system’s capacity to deliver liquid contents into the oral cavity, without handling the animal. Previous bait acceptance and uptake studies using a dye incorporated in the sachet found that it was extremely difficult to observe the coloration of an animal’s tongue and oral mucous membranes without direct observation, which required restraint [22,23,24]. The results of our study support the idea of using a more intense colorant, such as Patent Blue V, at the appropriate concentration, to ascertain release of sachet contents into the oral cavity of an animal that has been offered an oral rabies vaccine bait, without restraining it.

## 2. Materials and Methods

### 2.1. Animals and Housing

Our study group consisted of 47 adult dogs of mixed gender and age, kept for purposes of serology evaluation in unrelated studies at the study site, Bangkok Metropolitan Administration’s dog shelter in Taptan, Uthai-Thani province, Thailand [25]. These dogs had been separated from other shelter dogs for over a year prior to the start of our study. They had received a combination vaccine against canine distemper, parvovirus, adenovirus infection, bronchitis, and leptospirosis (RECOMBITEK^®^ C8, Merial (Thailand) Ltd., Bangkok, Thailand) at 2, 3 and 4 months old, respectively. As part of the aforementioned study, all dogs had been vaccinated against rabies and the presence of protective levels of rabies virus antibodies in their blood was confirmed. All people involved in the trials had received pre-exposure rabies prophylaxis as well.

Dogs were kept in pairs, except for few exceptions, where social behavior made caging with conspecifics impossible. The enclosures totaled 36 cages in two rows back-to-back, of which 26 cages held the dogs. (Table 1). Standard cage sizes were 90 cm × 350 cm × 190 cm (length × width × height). The floor and lower part of the wall (100 cm) were made of concrete. The upper part, as well as the ceiling, consisted of wire fencing. Each cage was accessible through an open fenced door, reinforced with extra wire fencing where necessary (Figure 1a,b).

Animals were fed once per day with commercial dog food (Blue’s canine special, BLUEFALO Co. Ltd., Nakhon Pathom, Thailand). For the duration of the study, feeding time was moved from around noon to the later hours of the day, after trials had finished. Water was offered at libitum. Cages were completely cleaned once per day.

### 2.2. Bait Types

The bait used in this study consisted of a sachet measuring 8 cm × 3 cm (length × width; Figure 2), made of a proprietary foil, dipped in an egg-flavored gelatin-mass (Ceva Santé Animale, internal data). Each sachet contained a volume of 3 mL. Four (4) types of sachet fillings were used. Bait type A contained the colorant Green S (Cas.Nr.: 3087-16-9, Acros -Organics BVBA, Thermo Fisher Scientific, Geel, Belgium) in a 50% sucrose solution (Cas.Nr.: 8013-17-0, Joh. Vögele KG, Lauffen a.N., Germany). Bait type B contained Patent Blue V dye (Cas.Nr.: 20262-76-4, Acros Organics BVBA, Thermo Fisher Scientific, Geel, Belgium) in a 50% sucrose solution. Both colorants are commonly used in food products. Dyes were added at 2.5% of the final filling volume. Both dyes and dosages used were selected based on previous screening studies (A. Langguth, unpublished results). Bait type C and D did not contain any dye. Type C contained a standard medium used for rabies vaccine production (Ceva Santé Animale, internal data). Type D contained denatonium benzoate (Bitrex^®^, Macfarlan Smith Ltd., Edinburgh, UK), typically used as a deterrent to prevent ingestion of toxic and otherwise harmful or undesirable materials in animals and children alike, due to its bitterness [26]. This was used as a solution of 10 ppm.

### 2.3. Study Design

The study was conducted from 28 January to 31 January 2020. Over this study period of 4 days, each dog was fed each type of bait once. The order in which the baits were fed was randomized (Appendix A). Trials began at 09:00 am Indochina Time (IDT) each day and ended once all the baits were fed to the dogs, with a one-hour break at noon. Trials usually ended at around 01:00 to 02:00 pm IDT.

### 2.4. Testing Procedure

Dogs were individually identified and offered the bait. The type of bait a dog would receive was determined by random assignment prior to the start of the trials. Reaction to the bait was evaluated using a standardized paper form. In cases where type A or B baits were fed, dogs were examined for traces of dye after bait consumption. Since there were 2 dogs caged together in most cases, one of the dogs was removed from the cage by a caretaker prior to offering the allocated bait to the caged dog. An observation was regarded as concluded when the dog discarded or swallowed the sachet or when the maximum observation time of 5 min (300 s) was reached. After the observation was concluded, the dog was taken out of the cage, examined, and exchanged for the dog first removed. Before offering the second bait to the second dog, any unconsumed bait remnants, including discarded blisters, were collected and the cage was hosed down fully, also removing any potential remnants of dye. After another observation period of 5 min maximum, leftover and discarded bait material was again collected, and the second dog was taken out of the cage for examination. The position of the dogs in the cages did not change throughout the study; however, the order in which the dogs were removed from the cages, and thus which dog was allowed to eat its bait first, was reversed on days 2 and 4, respectively. We recorded the time until first contact with the bait occurred, bait acceptance overall, total handling time, and number of chews. The latter was evaluated using a mechanical hand counter (Group Silverline Limited, Yeovil, UK). Perforation of the sachet was either determined by visual confirmation, when type A or B baits were fed, or by examination of bait remains, after collecting them from the cage. Ingestion of the sachet was noted as well. Visibility of dye from a distance of 5 m was noted; this included staining marks on the dog as well as visible spillage on the floor of the cage. For evaluation of dye capacities, photographs were taken of the left and right side of each dog’s face, as well as the front of the muzzle (Figure 3). The tongue and oral mucous membranes were also photographed. Intensity of staining was ranked I, II, and III and noted for all locations (I corresponding to no visible staining, II corresponding to weak staining, and III corresponding to clear staining) (Figure 4).

### 2.5. Statistical Analysis

Statistical analyses were carried out using GraphPad Prism v9.0 (GraphPad Software Inc., San Diego, CA, USA) and R v4.0.2 (The R Foundation for Statistical Computing, Vienna, Austria) [27]. The comparison of means for number of chews by bait type and study day were determined by a generalized least-squares (GLS) method using “nlme” package [28], followed by Tukey’s tests.

Various multivariate regression models were performed to assess the relationship between a dependent variable and independent variables. The regression models were based on a mixed model analysis that considered the individual dog as a random effect to adjust for clustering of repeated measures within dogs. Individual independent variables were initially screened for associations with a dependent variable, with a cutoff of *p* < 0.20, and eligible variables were used to build the multivariable model using the forward manual stepwise process with maximum likelihood test. Factors with *p* < 0.05 in the multivariate analysis were included into the final model. Independent variables were bait type (A, B, C, and D) and study day (1, 2, 3, and 4). Dependent variables were bait acceptance (0—no, 1—yes), sachet discarded (0—no, 1—yes), visibility at 5 m (0—no, 1—yes), staining of the tongue (0—no staining visible, 1—weak and clear staining visible), as well as staining of the oral mucosa (0—no staining visible, 1—weak and clear staining visible). A generalized linear mixed-effects model (GLMM) method was conducted using the “lme4” package [29]. A linear mixed-effects model (LMMs) using “nlme” package [28] was produced to determine the association of independent variables and number of chews as the dependent variable. 

Bait handling time was estimated using Kaplan–Meier survival curves, accounting for censored observations. Two time-to-event variables, namely, time to first contact with the bait (measured in seconds; from offering the bait to first contact with the bait) and bait handling time (measured in seconds; from first contact with the bait to either consumption or discarding of the bait), were used as dependent variables for the survival analysis. A duration (both time to first contact and handling time) greater than 300 s was defined as censored in this analysis. Log-rank test and Cox’s proportional hazard model using the “survival” package, according to Therneau and Grambsch [30], were performed to determine potential differences in the Kaplan–Meier survival function of bait handling time by bait type and study day. The effects of the independent variables on the time until first contact with the bait occurred, as well as bait handling time, were examined with a Cox model containing mixed (random and fixed) effects using the “coxme” package [31].

## 3. Results

### 3.1. Initial Reaction to the Bait

Overall, bait acceptance was very high. In 91.5% of total observations, dogs accepted the bait (Table 2). Sixteen (16) baits offered were not accepted by a total of 6 different dogs; 2 dogs refused a single bait, 1 dog refused 2 baits, and 3 dogs did not accept any of the 4 baits offered.

Bait type B had the highest acceptance rate overall, with 93.8%. However, bait acceptance did not differ markedly between bait types. Acceptance for bait type D, the negative control, was not notably lower than for other bait types.

Accidentally, 6 dogs received a certain bait type twice and consequently were not offered one particular bait type: dog 36—AACD, dog 39—DDBA, dog 40—AABD, dog 42—BACC, dog 43—BBDC and dog 46—CDBB. Bait acceptance was slightly lower on the first study day (87.2%) compared to the following three study days (91.5–93.6%) (Table 3). Of the animals ultimately consuming the bait, more dogs on the first study day (48.8%) did not immediately make contact with the baits upon offering than on subsequent study days (18.6–20.5%).

### 3.2. Number of Chews and Handling Time

We examined the influence of bait type, as well as study day, on number of chews and bait handling time (capped at 300 s). The differences in means of number of chews between the different bait types and study days are shown in Table 4. Bait type A had the lowest average number of chews. Meanwhile, the first study day had the highest average number of chews.

The final model of factors significantly associated with the number of chews is shown in Table 5. The association between the number of chews and study day was negative; for bait type it was positive. For days 2, 3, and 4, a significantly decreased number of chews compared to day 1 was observed. An increased number of chews was documented for bait types B, C, and D compared to type A.

A Kaplan–Meier curve on bait handling time among bait types is shown in Figure 5. Median (and 95% CI) of the Kaplan–Meier curve indicated that 50% of bait type A, B, C, and D were, after first contact, either consumed or discarded at 80 (60–156), 118 (105–165), 85 (65–130), and 110 (73–150) seconds, respectively. Comparing bait handling time between the different types of bait using a log-rank test, there was no significant difference found (*p* = 0.7). A Kaplan–Meier curve on bait handling time according to study day is shown in Figure 6. Median (and 95% CI) of the Kaplan–Meier curve indicated that 50% of bait on days 1, 2, 3, and 4 were either consumed or discarded after first contact at 130 (115–198), 88 (64–146), 111.5 (84–168), and 65.5 (53–106) seconds, respectively. Based on results from the log-rank test, there was a significant difference between bait handling time among certain study days (*p* = 0.002). The results of the Cox proportional hazard model showed that baits were handled significantly shorter on the 4th day of the study than on the first day of trials (*p* < 0.001; hazard ratio = 2.36).

Study day was significantly correlated with time until first contact with the bait was observed, as well as total bait handling time (Table 6). From the 2nd to the 4th study day, a significantly shorter time until first contact with the bait was observed as well as a significantly shorter total handling time in comparison to the first day.

### 3.3. Sachet Handling

The fate of the sachet is shown in Table 7. Bait type A was swallowed slightly less often than other bait types; Chi^2^-test, Chi^2^ = 7.88, df = 3, *p* = 0.05. The type of bait played an important role in the assessment of perforation. In cases where the sachet did not contain any dye and could not be recovered due to ingestion by the animal, it usually proved impossible to ascertain whether the sachet had been perforated or not. For 36.9% of the dogs offered bait types C and D, it could thus not be determined if sachet contents were released in the oral cavity or not. Meanwhile, determination of sachet perforation was possible in all observations where dogs were offered bait type A or B. No significant difference in handling between bait types A and B was observed; Fisher’s exact test, *p* = 0.15.

### 3.4. Mixed-Effects Binomial Regression Model

The final models of factors associated with bait acceptance, discarding of the sachet, visibility at 5 m, staining of the tongue, and staining of the oral mucous membranes are shown in Table 8. Study day was significantly associated with bait acceptance, discarding of the sachet and staining of the tongue. Bait type was significantly associated with discarding of the sachet, visibility at 5 m, staining of the tongue, and staining of the oral mucous membranes. Baits were overall significantly less accepted on the first day of trials. On the 3rd and 4th study day, bait was significantly more often discarded rather than swallowed. Bait type C and D were associated with a decreased observation of discarded bait compared to type A. Overall, staining of the tongue was observed significantly less during the 3rd and 4th study days. Bait type B was significantly better visible from a distance of 5 m than type A. Bait type B also had a significantly higher chance of staining the tongue and oral mucosa, compared to bait type A.

## 4. Discussion

Oral rabies vaccination has been shown to be a powerful tool to control and ultimately eliminate rabies from targeted reservoir species in different settings around the world [32,33,34,35]. It is suggested that ORV can contribute significantly to the elimination of dog-mediated rabies in areas with a high number of animals inaccessible for parenteral vaccination [36]. One of the pre-requisites for ORV is a suitable bait. Many bait studies have been conducted all over the world to evaluate bait palatability and acceptance in dogs [23,24,37,38,39,40,41,42], but only few have directly addressed vaccine release in the oral cavity [22].

This study gives detailed insights into the handling of a highly attractive bait by dogs. The use of oral color indicators in ORV baits, as surrogates of live oral rabies vaccines, provides the ability to determine successful release of sachet contents and thus allows evaluation of the efficacy of newly developed bait candidates.

In this study, the dog’s response to baits containing the medium used in the vaccine formulation was assumed to be the same as the response to baits filled with actual oral rabies vaccine. Since bait handling did not differ significantly between bait types A, B, and C, it can be concluded that dogs would manipulate a bait filled with oral rabies vaccine similar to the way handling was observed in type A and B baits. Bait type D, containing denatonium benzoate, was considered a negative control. However, bait acceptance and handling time of bait type D did mostly not differ from the other bait types. While these results suggest that bait filling cannot be detected through the sachet and coating by dogs, they also underscore the fact that perception of smell and taste are not necessarily identical among species. Denatonium benzoate has been used as an aversive agent for rodents [43], but since no studies on the use of denatonium benzoate in dogs exist to date, it cannot be ruled out that the concentration used in our studies was too low for dogs to detect bitterness as intended. 

A promising result is the fact that, even though dogs were unfamiliar with the baits offered to them, acceptance rate was generally high. However, it should be noted that bait acceptance and handling on the first day differed from subsequent days. On day 1 of the study, not only was bait acceptance lower (Table 3), but the animals also chewed the baits for a longer period of time (Figure 6). On the 4th day, a significantly shorter handling time was observed, but this did not differ between bait types. Overall, the sachet was more often discarded during the last days of the study (Table 8). The fact that baits were manipulated less by dogs on the 3rd and 4th days of the study also explains why less staining was generally observed on these last two test days. In summary, these findings suggest that there is a certain effect of conditioning to newly offered bait. While dogs were hesitant to approach the bait at first, they started to develop more effective methods to consume only the bait mass itself. This is evidenced in the increase in discarding of the sachet, as swallowing allows the dog to consume as much of the bait mass as possible if they are initially unable to separate it from the sachet. While this conditioning effect has led to better acceptance rates and a shorter handling time, it may also hinder effective bait delivery during single-day ORV campaigns. Still, overall bait acceptance was 91.5% (Table 2), implying that even in populations wholly unfamiliar with oral rabies vaccine baits, sufficient vaccination coverage can be achieved. Pre-baiting with placebo baits prior to the actual campaign may improve bait acceptance overall. However, these improvements may only be marginal and would have to be evaluated for cost-effectiveness. Furthermore, since animals participating in this study were being kept for other study purposes and were therefore accustomed to human interaction and observation, these results could differ under field settings. Nevertheless, study results on free-roaming dogs showed an acceptance rate of 92.8% when offered an identical egg bait, which they had also never encountered before [44].

Compared to other bait types, type A was swallowed less often (Table 7). One possible explanation for this might be the bitter taste of the dye itself. Animals were observed trying to separate the sachet from the bait matrix, instead of continuing to chew the bait after perforating it. In many cases, this was not completely possible, which may have discouraged animals from continuing to handle the bait.

Counterintuitively, coloring was also observed in some animals that had not perforated the sachet. Due to the manual filling of sachets, complete sealing of the dye injection site could not be guaranteed in all cases (S. Ortmann, 2021 personal communication, 8 April). Thus, small amounts of the liquid contents were able to leak out when subjected to pressure by dogs taking up the bait, without actual perforation of the sachet occurring.

Patent Blue V appears to be a promising dye for use as an oral marker substance. In the oral cavity, this dye was especially well visible on the tongue; most likely due to the tongue’s rougher surface compared to the oral mucosa. Although this same dye has been used in previous studies to evaluate bait acceptance, the staining in these studies was found to be less pronounced and was of limited use to evaluate vaccine release in the oral cavity [22,23,24]. This difference is most likely associated with the lower concentration of dye used.

Although visibility of the dye from a distance of 5 m was less pronounced than when examined at close range, the use of Patent Blue V at a concentration such as that used in this study offers many other potential applications. One of these addresses the extreme difficulty to estimate vaccination coverage obtained during oral vaccination campaigns. Per definition, the target population of ORV campaigns are animals that cannot be restrained and handled without special effort. Hence, there is no possibility to mark these dogs by methods such as collaring. Remote application of a marking substance is not easy and provides unreliable results [12,45]. In contrast, staining of the tongue through use of Patent Blue V is often pronounced enough to be easily discernible from a short distance and may remain so for several hours. During this study, it was noted that dogs tested earliest in the day (09:00 am IDT) and which had been fed type B baits still showed visible signs of oral staining after trials had concluded (02:00 pm IDT). A capture-recapture survey conducted on the same day, after a vaccination campaign using placebo vaccine baits containing marker dye, may thus provide the opportunity to approximate vaccination coverage of a free-roaming dog population obtained with ORV under field conditions.

Furthermore, these baits can be used to obtain data evaluating potential human contacts with orally vaccinated dogs, and thereby, vaccine contact. As all oral rabies vaccines are based on replication-competent human pathogens, human direct and indirect contact with viable vaccine virus should be prevented. Although hundreds of millions of oral rabies vaccine baits targeted at different wildlife species have been distributed, predominantly in Europe and North America [34,46,47,48], only two serious adverse events in humans have been reported so far [49,50]. Nevertheless, when baits are distributed, targeting free-roaming dogs, the chances that humans will come into contact with the vaccine virus is much higher than when baits are distributed in the context of oral vaccination of wildlife against rabies. Not only do free-roaming dogs and humans share the same environment, but there is also often direct contact between them. It has been shown that viable oral rabies vaccine released in the oral cavity of dogs can be re-isolated from saliva collected several hours after the vaccine bait was offered [51]. Hence, the possibility of vaccine virus transmission through dogs biting or licking humans remains. The persistence of visible staining by Patent Blue V is similar to the persistence of viable vaccine virus in the oral cavity after oral vaccination. Additionally, it can be assumed that the intensity of staining would correlate with the amount of vaccine virus released in the oral cavity. Hence, during a follow-up survey after distribution of bait containing the marker substance, it could be estimated how many people would have had contact with the contents of ORV bait, as the dye also stains clothes and human skin.

None of the dogs in this study showed any adverse reaction after consuming the multiple baits offered to them during the study period of 4 days. In cases where the sachets were swallowed, staff were able to retrieve remnants of egested material from enclosures the next day. This implies that repeat ingestion of the bait itself, were it to occur under field conditions, would most likely not pose a risk to dogs. It should also be noted that in some cases where type A or B baits were consumed, traces of dye were visible in feces the following day. Nevertheless, this was an unexpected finding. Since complete separation of dogs for the entire study period was neither planned nor spontaneously possible, effects of type A versus type B baits cannot be distinguished in this respect.

We were able to discern individual changes in behavioral patterns regarding bait acceptance and handling over the course of our study using the selected study design. However, the fact that dogs gained increasing familiarity with baits could also impact the results. While this has to be taken into consideration when evaluating the efficacy of the ORV bait used in our study in general, it does not change the fact that, when compared directly to each other, the marker substance Patent Blue V has been shown to have stronger dyeing capacities than Green S, without there being any prominent difference in handling of baits filled with Patent Blue V compared to those filled with ORV-medium alone.

## 5. Conclusions

The results of this study support the idea of using dyes as a marker substance to assess the release of ORV bait contents in the oral cavity of dogs. Patent Blue V as a 2.5% solution in 50% fructose was shown to be a reliable indicator of sachet perforation and subsequent release of its contents into the oral cavity, which can be observed remotely without the need to restrain the animal for evaluation. This non-invasive method may aid in the future development and improvement of strategies for oral rabies vaccination in stray dogs, contributing to the global goal of zero human dog-mediated rabies deaths by 2030.

## Figures and Tables

**Figure 1 viruses-13-01382-f001:**
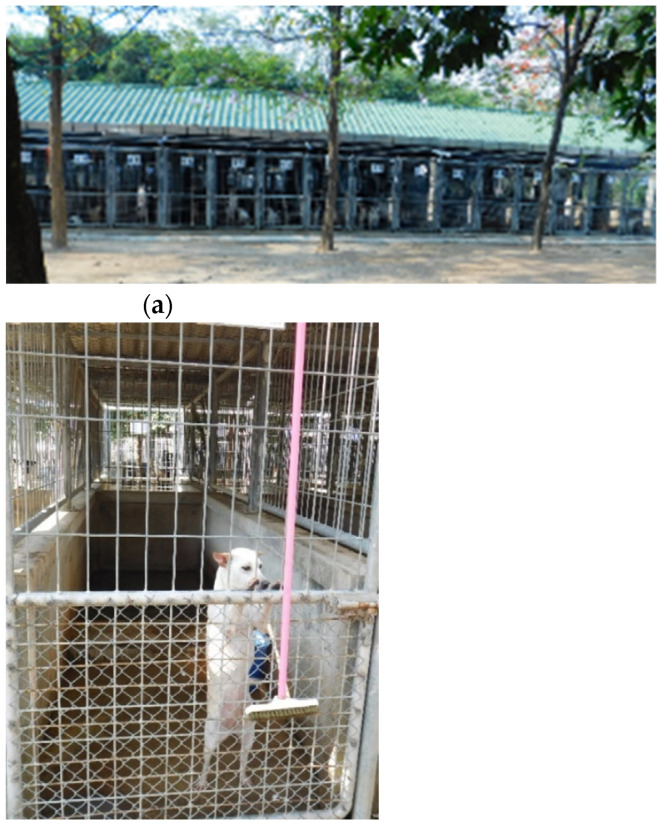
(**a**,**b**) Standard cages at the dog shelter in Taptan, Uthai-Thani province, Thailand.

**Figure 2 viruses-13-01382-f002:**
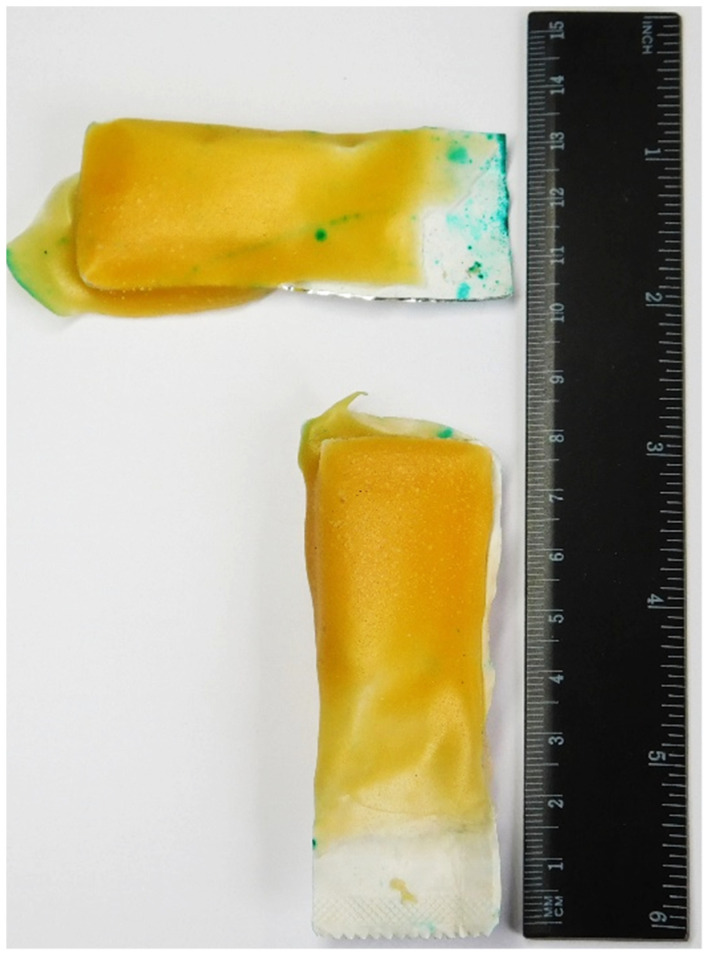
Bait measurements.

**Figure 3 viruses-13-01382-f003:**
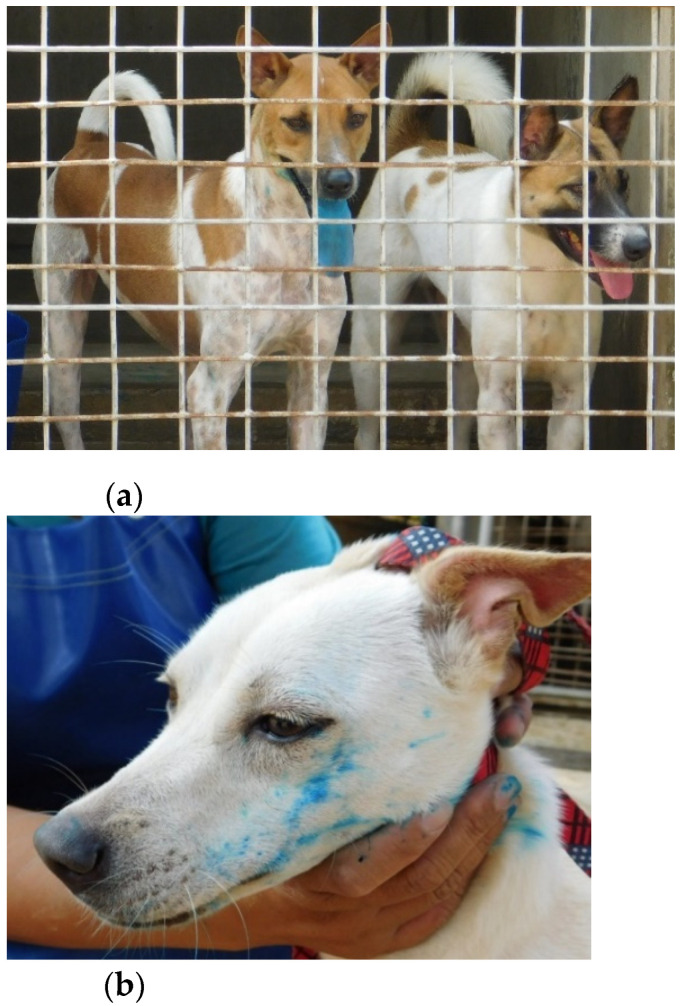
(**a**,**b**) Visibility of dye after bait consumption.

**Figure 4 viruses-13-01382-f004:**
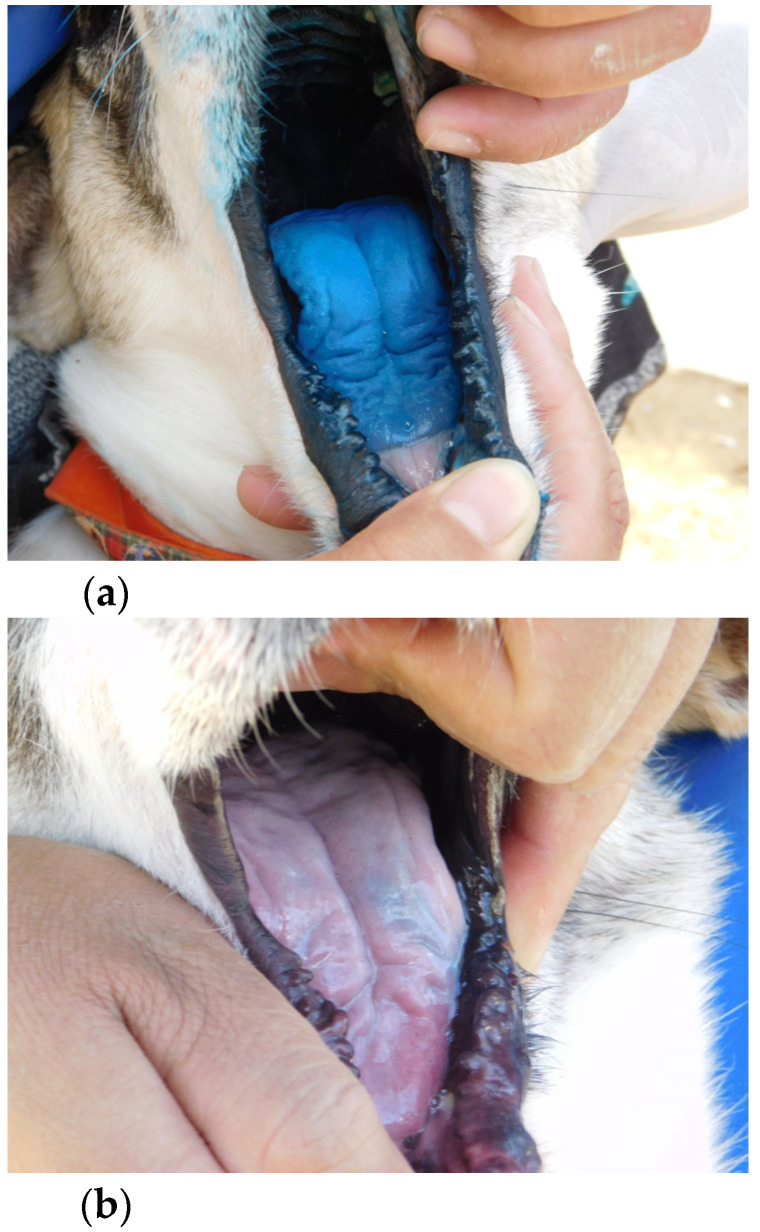
(**a**) Dog with intensive staining of the tongue (score III), indicating the release of a large volume of liquid dye in the oral cavity, compared to (**b**) another dog that showed hardly any staining (score II), indicating that only limited amounts of sachet contents were released in the oral cavity.

**Figure 5 viruses-13-01382-f005:**
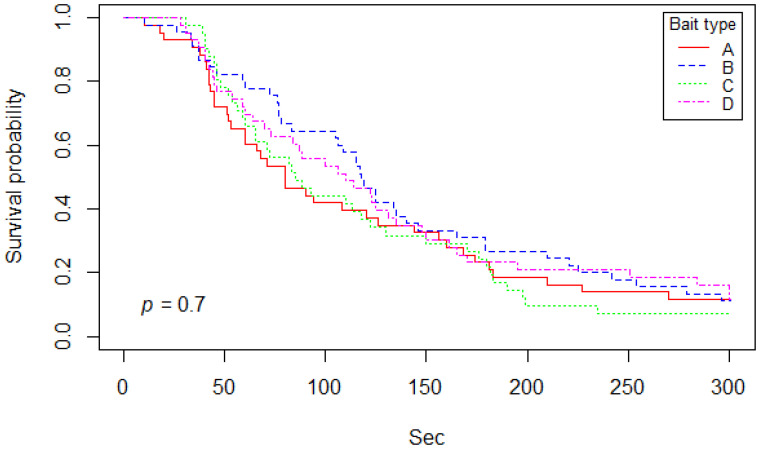
Kaplan–Meier curve of bait handling time separated based on bait type (A, B, C, and D).

**Figure 6 viruses-13-01382-f006:**
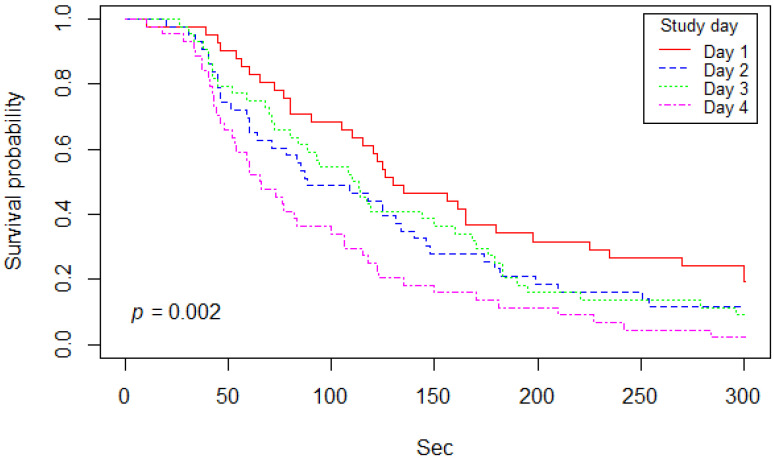
Kaplan–Meier curve of bait handling time separated based on study day (Day 1–28, day 2–29, day 3–30, and day 4–31 January 2020).

**Table 1 viruses-13-01382-t001:** Cage allocation.

Cage	1	2	3	4	5	6	7	8	9	10	11	12	13	14	15	16	17	18
Dog	1	3	5	7	9	11	13	15	17	19	21	23	25	27	29	31	33	34
	2	4	6	8	10	12	14	16	18	20	22	24	26	28	30	32		35
Dog	36	38	40	42	44	45	46	47										
	37	39	41	43														

**Table 2 viruses-13-01382-t002:** Overall bait acceptance according to bait type.

Bait Type	Bait Accepted		95% CI Accepted	
	No (%)	Yes (%)	Lower	Upper
A	4 (8.5)	43 (91.5)	81.6	97.0
B	3 (6.3)	45 (93.8)	84.6	98.3
C	5 (10.9)	41 (89.1)	78.5	96.6
D	4 (8.5)	43 (91.5)	81.6	97.0
total	16 (8.5)	172 (91.5)	87.4	94.6

**Table 3 viruses-13-01382-t003:** Overall bait acceptance according to study day.

Study Day	Bait Accepted		95% CI Accepted	
	No (%)	Yes (%)	Lower	Upper
1	6 (12.8)	41 (87.2)	76.4	94.3
2	4 (8.5)	43 (91.5)	81.2	97.0
3	3 (6.4)	44 (93.6)	84.3	98.2
4	3 (6.4)	44 (93.6)	84.3	98.2
total	16 (8.5)	172 (91.5)	87.4	94.6

**Table 4 viruses-13-01382-t004:** Mean ± SEM (Standard Error of Mean) of number of chews between the different bait types and study days in dogs.

Variables	Mean ± SEM	95% CI	*p*-Value
Bait type			<0.001
A	20.2 ± 4.87 ^a^	10.6–29.9	
B	42.0 ± 4.81 ^b^	32.5–51.5	
C	45.3 ± 5.14 ^b^	35.1–55.5	
D	50.7 ± 5.06 ^b^	40.7–60.7	
Study day			
1	65.1 ± 5.00 ^a^	55.0–75.2	<0.001
2	42.1 ± 4.98 ^b^	32.1–52.1	
3	28.9 ± 4.86 ^bc^	19.2–38.7	
4	22.0 ± 5.08 ^c^	11.8–32.2	

^a, b, c^ Values differ among variables; different letters indicate a significant difference between values (*p* < 0.01).

**Table 5 viruses-13-01382-t005:** Final linear mixed-effects model indicating factors associated with number of chews in dogs.

Dependent Variable	Factor	Coefficient	SE	95% CI	*p*-Value
Number of chews	Study day				
	1	reference			
	2	−24.75	6.85	−38.17–−11.33	<0.001
	3	−35.84	6.80	−49.16–−22.51	<0.001
	4	−44.76	6.81	−58.11–−31.41	<0.001
	Bait type				
	A	reference			
	B	21.83	6.68	8.74–34.93	<0.001
	C	25.03	6.84	11.62–38.44	<0.001
	D	29.22	6.79	15.90–42.54	<0.001

SE = Standard Error, 95%CI = 95% confidence interval.

**Table 6 viruses-13-01382-t006:** The final mixed effects Cox model indicating factors associated with time until first contact with the bait and bait handling time in dogs.

Dependent Variable	Factor	Coefficient	SE	HR	95% CI	*p*-Value
Time until first contact	Study day					
	1	reference				
	2	0.82	0.23	2.26	1.43–3.58	<0.001
	3	0.78	0.23	2.19	1.39–3.46	<0.001
	4	0.77	0.24	2.16	1.36–3.45	0.001
Handling time	Study day					
	1	reference				
	2	0.93	0.26	2.53	1.51–4.25	<0.001
	3	0.97	0.26	2.64	1.57–4.41	<0.001
	4	1.63	0.28	5.11	2.97–8.80	<0.001

SE = Standard Error, HR = Hazard ratio, 95%CI = 95% confidence interval

**Table 7 viruses-13-01382-t007:** Fate of the sachet in relation to bait type. Sachet discarded versus swallowed and sachet perforated versus not perforated.

Bait Type	Discarded (%)	Swallowed (%)	Sachet Perforated		
			Yes (%)	No (%)	Unknown (%)
A	35 (81.4)	8 (18.6)	37 (86.0)	6 (14.0)	0 (-)
B	30 (66.7)	15 (33.3)	43 (95.6)	2 (4.4)	0 (-)
C	22 (53.7)	19 (46.3)	20 (48.8)	3 (7.3)	18 (43.9)
D	26 (60.5)	17 (39.5)	25 (58.1)	5 (11.6)	13 (30.2)
total	113 (65.7)	59 (34.3)	125 (72.7)	16 (9.3)	31 (18.0)

**Table 8 viruses-13-01382-t008:** The final generalized linear mixed-effects model indicating factors associated with bait acceptance, sachet discarded, visibility at 5 m, staining of the tongue, and staining of the oral mucous membranes in dogs.

Dependent Variable	Factor	Coefficient	SE	OR	95% CI	*p*-Value
Bait acceptance	Study day					
	1	reference				
	2	15.12	0.004	3.70 × 10^6^	3.68 × 10^6^–3.73 × 10^6^	<0.001
	3	29.37	0.004	5.69 × 10^12^	5.64 × 10^12^–5.73 × 10^12^	<0.001
	4	29.59	0.004	7.11 × 10^12^	7.05 × 10^12^–7.17 × 10^12^	<0.001
Sachet discarded	Study day					
	1	reference				
	2	0.98	0.53	2.68	0.95–7.54	0.063
	3	1.10	0.54	3.01	1.05–8.61	0.040
	4	1.40	0.55	4.05	1.38–11.90	0.011
	Bait type					
	A	reference				
	B	−0.98	0.57	0.38	0.12–1.15	0.086
	C	−1.54	0.58	0.21	0.07–0.67	0.008
	D	−1.29	0.58	0.28	0.09–0.85	0.025
Visibility at 5 m	Bait type					
	A	reference				
	B	2.55	0.83	12.87	2.55–64.95	0.002
Staining, tongue	Study day					
	1	reference				
	2	−5.86	3.31	2.85 × 10^−3^	4.35 × 10^−6^–1.87 × 10^0^	0.077
	3	−17.97	1.58	1.56 × 10^−8^	7.05 × 10^−10^–3.46 × 10^−7^	<0.001
	4	−10.97	0.004	1.71 × 10^−5^	1.70 × 10^−5^–1.73 × 10^−5^	<0.001
	Bait type					
	A	reference				
	B	3.14	0.005	23.01	22.78–23.23	<0.001
Staining, tonsils	Bait type					
	A	reference				
	B	3.62	0.70	37.40	9.44–148.11	<0.001

SE = Standard Error, OR = Odds ratio, 95%CI = 95% confidence interval.

## Data Availability

The datasets used and analyzed during the current study are available from the corresponding author on reasonable request.

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
