# Peer review of "Comparative Study of Optical Markers to Assess Bait System Efficiency Concerning Vaccine Release in the Oral Cavity of Dogs"

_viruses, 2021, doi:10.3390/v13071382_

Round 1
Reviewer 1 Report
This article is very well written. To fight against rabies in dog populations, according to the context and the location, ORV can be an useful tool as alternative method to the mass parenteral vaccination or as complement method to reinforce it. The study aimed to find a non-invasive method to assess the vaccine release in the dog oral cavity after bait ingestion allowing to estimate the vaccination coverage. The authors tested two dyes as markers i.e. Green S or Patent V Blue diluted in a fructose solution. According to this study, the most interesting results were obtained with the Patent V Blue. By using dyes, the vaccination coverage after the ORV campaign, could be rapidly, visually and safely assessed. In this paper, the authors underlined the numerous advantages to use dyes in oral baits as marker of the release of the vaccine in the oral cavity, including the evaluation of potential human contacts with the attenuated vaccine contained in the baits or with orally vaccinated dogs (licking or biting).
This paper is of great importance and shall be shared with the international scientific community involving in the fight of rabies in dogs. Moreover, this study was in line with the One Health concept “Zero by 30: The Global Strategic Plan to Prevent Human Deaths from Dog-Transmitted Rabies by 2030”
2.4 Testing procedure
This part should be rewritten to be clearer for the reader. This part raises some questions:
- Is the bait offered one per one to the dog?
- Is there a change of the dog in the cage after each tested bait or only after testing the 4 baits?
- Is the observation period of 5 min done for each bait, if they are given individually?
- Is the cage cleaning before the exchange of dog (not only the discarding of blisters) in case of dye remained on the floor?
Author Response
Paper ID - viruses-1274377
04/07/2021
Author's Reply to the Review Report
Anna Langguth
Author’s replies have been added in blue.
Point 1: 2.4 Testing Procedure
This part should be rewritten to be clearer for the reader. This part raises some questions:
Response 1: Thank you for your review and your questions about this part of our methodology; this was very helpful in identifying where there were shortcomings in our descriptions.
Point 2: Is the bait offered one per one to the dog?’
Response 2: Yes, it is. The type of bait was determined through random allocation before trials started. During trials, dogs were individually identified and offered the specific type of bait that had been chosen for them. Text will be adapted accordingly to make it clearer for the reader.
Point 3: Is there a change of the dog in the cage after each tested bait or only after testing the 4 baits?
Response 3: Two dogs were caged together in most cases. To ensure that the dog about to be tested would only eat the bait selected for them, the other dog was removed from the cage. Once the first dog had consumed the bait or an observation period of 5 minutes maximum was reached, the observation was concluded, and dogs were exchanged for one another. After observing the second dog in the same way, we switched to the next cage and repeated this procedure. The position of the dogs in the cages did not change throughout the study; however, the order in which the dogs were removed from the cages - i.e., which dog was allowed to eat its bait first - was reversed on days 2 and 4, respectively. These details will be added to the manuscript.
Point 4: Is the observation period of 5 min done for each bait, if they are given individually?
Response 4: Yes. Not all observations resulted in a maximum observation period of 5 minutes though, because an observation was also noted as “concluded” if dogs consumed or discarded the bait. We will add this detail.
Point 5: Is the cage cleaning before the exchange of dog (not only the discarding of blisters) in case of dye remained on the floor?
Response 5: Cages were cleaned after the first dog was removed from the cage and before the second dog was moved into it. Dye residue on the cage floor was easily removed by cleaning the concrete with a water hose. We will add this fact in our description of the methods used.

Reviewer 2 Report
The aim of this study is to compare utilities of two food dyes, Green S and Patent Blue V, as a marker substance contained in baits for oral rabies vaccination for dogs, which enables to evaluate vaccine release in the oral cavity. For this aim, a total of 47 shelter dogs in Thailand were randomly given baits containing one of the dyes, and then their behaviors such as bait acceptance, number of chews, and bait handling time were monitored. Also, efficiencies of dye staining in dogs were compared. Based on the findings obtained in this study, the authors conclude that Patent Blue V is suitable as a marker for evaluation of vaccine release in the oral cavity.
This is a well-written paper, providing basic information, which is very important for development of bait rabies vaccines for dogs in the future. I have only a few comments on this manuscript.
Major comments
- A total of 47 dogs were randomly offered one of four bait types (A-D) every day during the study period of 4 days. Many data obtained in this study indicate that learned behaviors of dogs, which probably results from the repeated bait offering, clearly affect the outcomes of this study. This would be the biggest weakness of this study. So I suggest that the authors should explain why they chose this study design in more detail and also should address the advantages and disadvantages of this study setting in Discussion.
Minor comments
- “discoloration”(line 153): This may cause misunderstanding. “staining” would be more appropriate.
- “Study day 1-4”are shown as “Study day 28-31”in Figure 5 and Table 7. To avoid confusion, it is desirable to unify the indication: “Study day 1-4” would be better than “Study day 28-31.”
- “more often”(line 263): This must be mistake for “less often.”
- In Discussion section, please refer Figures and Tables that contain the data being discussed.
Author Response
Paper ID - viruses-1274377
04/07/2021
Author's Reply to the Review Report
Anna Langguth
Author’s replies have been added in blue.
Major comments
Point 1: A total of 47 dogs were randomly offered one of four bait types (A-D) every day during the study period of 4 days. Many data obtained in this study indicate that learned behaviors of dogs, which probably results from the repeated bait offering, clearly affect the outcomes of this study. This would be the biggest weakness of this study. So I suggest that the authors should explain why they chose this study design in more detail and also should address the advantages and disadvantages of this study setting in Discussion.
Response 1: This is a very important point. Thank you for bringing it up.
We chose this study design partly because of practicality and partly because it allowed us to examine the effect that repeat bait offering would have on bait acceptance and handling.
The 47 dogs participating in our study were pre-selected for an unrelated serology study and had thus already been separated from the rest of the other shelter dogs, making them easily accessible. This also meant that they were used to regular handling, making trials less stressful for the animals and less work-intensive in general, requiring less personnel.
We will include a more in-depth discussion of our study design.
Minor comments
Point 2: “discoloration”(line 153): This may cause misunderstanding. “staining” would be more appropriate.
Response 2: Agreed. This will be changed.
Point 3: “Study day 1-4”are shown as “Study day 28-31”in Figure 5 and Table 7. To avoid confusion, it is desirable to unify the indication: “Study day 1-4” would be better than “Study day 28-31.”
Response 3: This will also be changed; thank you for pointing it out.
Point 4: “more often”(line 263): This must be mistake for “less often.”
Response 4: Very true. We will make sure results and discussion will be changed accordingly.
Point 5: In Discussion section, please refer Figures and Tables that contain the data being discussed.
Response 5: We will include these references.

Reviewer 3 Report
Langguth and co-authors evaluated using a dye in oral rabies vaccine baits as an indicator of vaccine up-take by dogs. I have no major concerns with the manuscript.
Minor comments
Line 123: I could not find an in-line citation for figure 2. Please add a citation or remove the figure.
Line 158: I have concerns about photos of ungloved hands inside the dogs mouth. While I agree that PPE is not necessary to handle apparently healthy dogs some more details in the methods section would lessen my concern. Given Thailand still has canine rabies how long were the dogs quarantined before being co-housed, time in captivity before your study started, and was a health assessment completed before starting the study. Was a risk assessment completed to determine if PPE was needed?
Line 208: appears to be misplaced inside the table
Line 317: none of the conclusions relied on the dogs rejecting type D. Why include this group in the final analysis?
Line 384: I think the correct citation is 48-49 not 44-45. Also only 46 references are cited in the text but 50 are listed in the bibliography. I would check all of the in-line citation to ensure they are correct since at least this one appears out of order.
Author Response
Paper ID - viruses-1274377
04/07/2021
Author's Reply to the Review Report
Anna Langguth
Author’s replies have been added in blue.
Point 1: Line 123: I could not find an in-line citation for figure 2. Please add a citation or remove the figure.
Response 1: Figure 2 has been cited in line 112; shortly after describing the bait sachet.
Point 2: Line 158: I have concerns about photos of ungloved hands inside the dogs mouth. While I agree that PPE is not necessary to handle apparently healthy dogs some more details in the methods section would lessen my concern. Given Thailand still has canine rabies how long were the dogs quarantined before being co-housed, time in captivity before your study started, and was a health assessment completed before starting the study. Was a risk assessment completed to determine if PPE was needed?
Response 2: Thank you, that is indeed a very important point. For the most part, the dogs had been in the shelter since birth, separated from other shelter dogs for over a year prior to the start of our study. A baseline health examination had been performed as part of the referenced serologic study, and all dogs had been vaccinated against rabies. All personnel involved had also received pre-exposure prophylaxis against rabies. We will provide these details in our methodology section.
Point 3: Line 208: appears to be misplaced inside the table
Response 3: I have checked this, but I am unsure what appears to be misplaced inside the table. Could you please specify?
Point 4: Line 317: none of the conclusions relied on the dogs rejecting type D. Why include this group in the final analysis?
Response 4: We included type D in the final analysis to be able to compare all available data. Although denatonium benzoate did not cause as negative a reaction as we had hoped, the fact that this bitter substance did not prevent the dogs from eating the baits is something that can be compared to the results we obtained when using the dyes; since these are also bitter substances.
Point 5: Line 384: I think the correct citation is 48-49 not 44-45. Also only 46 references are cited in the text but 50 are listed in the bibliography. I would check all of the in-line citation to ensure they are correct since at least this one appears out of order.
Response 5: Thank you for this very helpful observation; we will go over all the in-line citations again to make sure all of them have been used correctly.
